# Robust Fine-Tuning from Non-Robust Pretrained Models: Mitigating Suboptimal Transfer With Epsilon-Scheduling

## Abstract

Fine-tuning pretrained models is the standard approach in current machine learning practice, but simultaneously achieving adversarial robustness to adversarial examples remains a challenge. Despite the abundance of non-robust pretrained models in open-source repositories, their use for Robust Fine-Tuning (RFT) remains understudied. This work aims to bridge this knowledge gap by systematically examining RFT from such models. Our experiments reveal that fine-tuning non-robust models with a robust objective, even under small perturbations, can lead to poor performance, a phenomenon that we dub *suboptimal transfer*. In fact, we find that fine-tuning using a robust objective impedes task alignment at the beginning of training and eventually prevents optimal transfer. To promote optimal transfer, we propose *Epsilon-Scheduling*, a simple heuristic scheduling over perturbation strength. Additionally, we introduce *expected robustness*, a metric that measures performance across a range of perturbations. Experiments on six pretrained models and five datasets show that *Epsilon-Scheduling* prevents *suboptimal transfer* and consistently improves the expected robustness.

## 1 Introduction

Fine-tuning pretrained models is the standard workflow in machine learning, spanning NLP (Koroteev, 2021) and vision (Goldblum et al., 2023). This workflow offers clear benefits: (i) reusing a single foundation model across tasks (Devlin et al., 2019), (ii) faster convergence and better generalization than training from scratch (Yosinski et al., 2014), and (iii) reduced computation (Weiss et al., 2016), especially when labelled data is scarce (Pan & Yang, 2010).

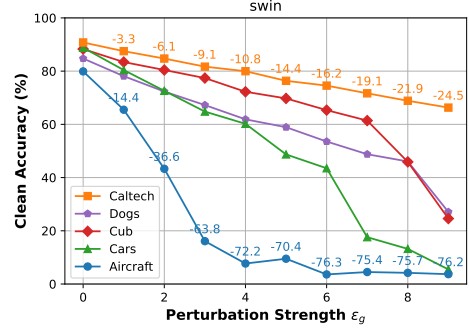

Figure 1: Robust Fine-Tuning can lead to *suboptimal transfer* even when optimizing for small perturbations.

However, in high-stakes applications, adversarial vulnerability remains a major concern (Biggio et al., 2013; Goodfellow et al., 2015). Adversarial Training (AT) (Madry et al., 2018) and its variants (Zhang et al., 2019; Wang et al., 2020; Ding et al., 2020; Shafahi et al., 2019a; Wong et al., 2020) are the most successful empirical defenses (Croce et al., 2020). Robust Fine-Tuning (RFT) is the integration of these methods in fine-tuning on downstream tasks (Shafahi et al., 2019b; Liu et al., 2023; Xu et al., 2024; Hua et al., 2024). RFT is challenging because it must balance alignment with the downstream task and robustness (Xu et al., 2024). Prior work mainly studies RFT from robust pretrained models (Hua et al., 2024; Liu et al.,

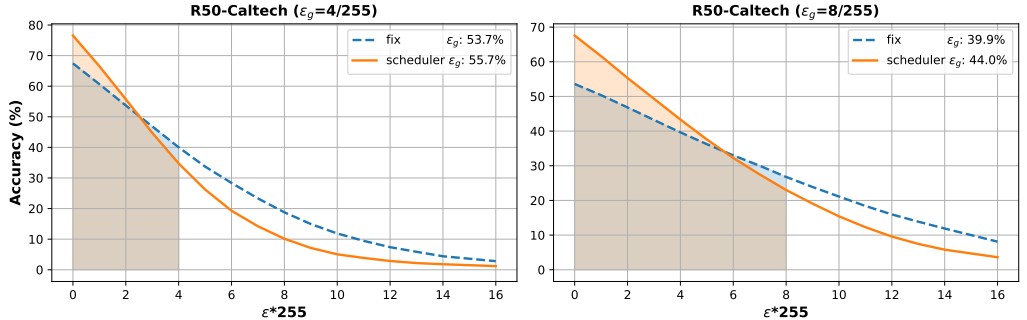

Figure 2: **Expected robustness**. The first value in the legend represents the evaluation over $[0, \epsilon_g]$ (shaded region) and the second value is over the whole interval $[0, 16]$.

2023; Xu et al., 2024), overlooking the more common non-robust ones (Wolf et al., 2020). Since robust models are costly and since pretraining typically targets general-purpose features, robustness can be considered as a property to be acquired on downstream tasks (Heuillet et al., 2025). Thus, improving RFT from non-robust backbones is essential and naturally aligns with current workflows.

In this work, we study Robust Fine-Tuning (RFT) of non-robustly pretrained backbones. We fine-tune various pretrained backbones on different datasets using adversarial training (Madry et al., 2018) with a fixed perturbation radius. We find that, even for small nonzero radii, this approach yields *suboptimal transfer*, where performance falls short of that achieved by standard fine-tuning (without perturbation) and is often too low to be considered a successful transfer. Its severity depends on both the backbone and the downstream task. Unlike standard fine-tuning, where model adaptation to the downstream task occurs immediately, our study shows that in RFT, **task alignment is delayed until later epochs**, eventually leading to *suboptimal transfer*.

To mitigate this, we propose ***Epsilon-Scheduling*** (Figure 3): a simple scheduling that starts with standard fine-tuning (zero perturbation) for early epochs and linearly increases to the target perturbation at final epochs. This strategy prevents *suboptimal transfer* and improves both generalization and robustness.

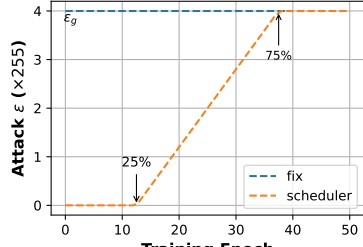

Finally, to better evaluate the fine-tuned models, instead of the standard evaluation that compares only clean and robust accuracy at target perturbation strength $\epsilon_g$, we introduce ***expected robustness***, which evaluates the expectation of the accuracy of the model across the full perturbation range $[0, \epsilon_g]$. In partic-

Figure 3: *Epsilon-Scheduling*

ular, cases where an increase in generalization comes at the cost of reduced robustness can make model comparison subjective. The *expected robustness* provides a comprehensive evaluation of the accuracy-robustness trade-off. Under this metric, *Epsilon-Scheduling* consistently improves performance, even when worst-case robustness at $\epsilon_g$ is similar or lower.

## 2 Methodology

**Robust Fine-Tuning**   Fine-tuning consists in training a classifier $f = c_{\theta_2} \circ h_{\theta_1}$, composed of a pretrained backbone $h_{\theta_1}$ and a randomly initialized classifier head $c_{\theta_2}$, to maximize accuracy on a given data distribution $\mathcal{D}$. This work focuses on full fine-tuning where both $\theta_1$ and $\theta_2$ are trainable parameters. In Robust Fine-Tuning (RFT), the goal is to maximize robust accuracy $\text{Acc}_{\epsilon_g}(f)$ at a target perturbation strength $\epsilon_g > 0$. We consider RFT with adversarial training (Madry et al., 2018) that minimizes the adversarial risk at $\epsilon$ as a surrogate objective:

$$R_\epsilon(f) = \mathbb{E}_{(x,y)\sim\mathcal{D}}\Big( \max_{\|\delta\|_\infty < \epsilon} \ell_{\text{CE}}(f(x + \delta), y) \Big) \tag{1}$$

where $\ell_{CE}$ the cross-entropy loss. The common practice in RFT for target perturbation strength $\epsilon_g$ consists of minimizing an empirical counterpart of $R_{\epsilon_g}(f)$ for a certain number of epochs, a strategy

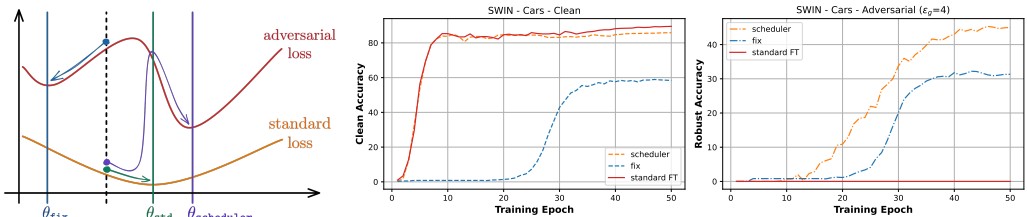

Figure 4: *Epsilon-Scheduling* **preserves task alignment while improving robustness.** Left: Illustrative example of the difference between RFT-fix and RFT-scheduler. Center and right: Evolution of clean and robust accuracy during the fine-tuning of the SWIN backbone on Cars dataset with $\epsilon_g = {}^4\!/_{255}$.

that we refer to as RFT-fix (or fix), since the training objective remains the same during the whole fine-tuning process.

***Epsilon-Scheduling*** In contrast with RFT-fix, we propose to achieve RFT for target perturbation strength $\epsilon_g$ by minimizing an empirical counterpart of $R_\epsilon(f)$ where the radius $\epsilon$ follows a simple schedule during the fine-tuning, as illustrated in (Figure 3). In effect, this strategy starts with standard fine-tuning at $\epsilon = 0$ for 25% of the number of epochs, then linearly increases from $\epsilon = 0$ to $\epsilon = \epsilon_g$ during half of the fine-tuning, until it finally minimizes $R_{\epsilon_g}(f)$ for the remaining 25%. From a transfer learning perspective, we can view this strategy as follows: begin with task adaptation, then gradually shift to the robust objective and conclude by minimizing the robust objective. In the sequel, we will refer to this strategy as RFT-scheduler (or scheduler).

**Expected Robustness** While RFT targets low adversarial risk $R_{\epsilon_g}(f)$, models are usually evaluated both for clean accuracy $\mathrm{Acc}_0(f)$ and robust accuracy $\mathrm{Acc}_{\epsilon_g}(f)$. We propose to extend this classical evaluation to take into account intermediary perturbation strengths within the range $[0, \epsilon_g]$. Evaluating models' accuracy at intermediate perturbation strengths reveals distinct patterns (See Figure 2). Such evaluation is helpful for comparing models with similar accuracies or when the clean–robust trade-off is ambiguous. We summarize these evaluations using the *expected robustness* metric, defined as the expectation under uniform distribution $U$ of the accuracy over $[0, \epsilon_g]$:

$$\mathbb{E}_{\epsilon \sim U[0,\epsilon_g]}\big[\mathrm{Acc}_\epsilon(f)\big] = \frac{1}{\epsilon_g}\int_0^{\epsilon_g} \mathrm{Acc}_\epsilon(f)\, d\epsilon = \frac{1}{\epsilon_g}\mathrm{AUC}_{\epsilon_g}(f)$$

where $\mathrm{AUC}_{\epsilon_g}(f)$ represents the area under the accuracy curve from 0 to $\epsilon_g$ (See Figure 2). More details can be found in Appendix B.

## 3 Characterizing Suboptimal Transfer in Robust Fine-Tuning

We explore how high perturbation strength $\epsilon_g$ in RFT-fix affects the transfer accuracy of non-robust pre-trained models. For our experiment, we use two ImageNet-pretrained backbones, SWIN and ViT, and fine-tune them on five datasets: Caltech256, Cub200, Stanford Dogs, Stanford Cars, and FGVC-Aircraft. We consider perturbation strengths $\epsilon_g$ from 0 (standard fine-tuning) up to ${}^9\!/_{255}$. The results in Figure 1 show that as $\epsilon_g$ increases, the transfer accuracy drops significantly. For example, at $\epsilon_g = {}^4\!/_{255}$, the SWIN models have performance drops of 10% to 72%, respectively, compared to standard fine-tuning. We refer to this phenomenon as ***suboptimal transfer***, where RFT-fix yields a transfer accuracy significantly

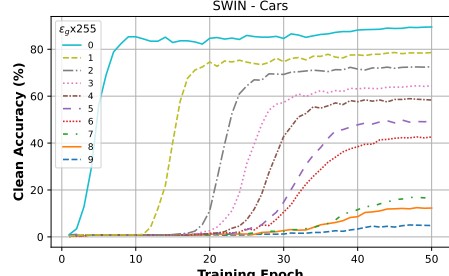

Figure 5: **RFT-fix delays task alignment**. The stronger the perturbation, the later the validation accuracy starts to improve.

lower than standard fine-tuning, at times to the point of no longer being considered an effective transfer. Results for ViT are in appendix (Figure 7)

**Robust Fine-Tuning with Fixed Perturbation Strength Delays Task Alignment** As shown in Figure 5 in standard fine-tuning, the task adaptation to the downstream task begins almost immediately–validation accuracy rises from the first epoch–since there are no robustness constraints that may conflict with task alignment. With nonzero values of $\epsilon_g$, RFT-fix distorts task-relevant features, which prevents early alignment and delays the onset of task adaptation. For example, task alignment

| Model | Setting | Aircraft Clean | Aircraft Adv. | Aircraft E. Adv. | Caltech Clean | Caltech Adv. | Caltech E. Adv. | Cars Clean | Cars Adv. | Cars E. Adv. | Cub Clean | Cub Adv. | Cub E. Adv. | Dogs Clean | Dogs Adv. | Dogs E. Adv. |
|---|---|---|---|---|---|---|---|---|---|---|---|---|---|---|---|---|
| vit | fix | 6.40 | 2.80 | 4.48 | 68.14 | 41.64 | 55.07 | 12.70 | 4.90 | 8.20 | 42.82 | 15.12 | 27.79 | 56.40 | **19.97** | 36.93 |
| | scheduler | **58.60** | **13.20** | **34.95** | **78.73** | **41.69** | **60.71** | **73.40** | **19.10** | **46.71** | **73.40** | **23.63** | **48.09** | **70.69** | 15.69 | **41.62** |
| swin | fix | 7.70 | 4.80 | 6.11 | 79.97 | **57.16** | 69.19 | 60.20 | 29.70 | 44.74 | 72.25 | **41.87** | 57.55 | 61.89 | **26.89** | 44.17 |
| | scheduler | **73.80** | **32.00** | **53.75** | **85.43** | 56.39 | **72.04** | **84.70** | **43.20** | **66.41** | **82.29** | 41.61 | **63.82** | **72.70** | 24.32 | **48.50** |
| convnext | fix | 7.60 | 4.50 | 5.86 | 83.27 | **61.54** | 73.08 | 69.60 | 43.20 | 57.52 | 76.34 | **47.08** | 62.59 | 68.90 | **31.61** | 50.61 |
| | scheduler | **78.40** | **38.00** | **59.40** | **89.41** | 61.45 | **77.23** | **88.90** | **57.70** | **75.85** | **85.17** | 44.99 | **67.30** | **78.39** | 26.31 | **53.19** |
| r50 | fix | 8.40 | 2.90 | 4.56 | 67.47 | **40.02** | 53.74 | 4.20 | 2.90 | 3.49 | 49.19 | 19.35 | 33.58 | 57.05 | **19.80** | 37.73 |
| | scheduler | **53.10** | **11.10** | **29.40** | **76.55** | 34.74 | **55.67** | **70.00** | **19.30** | **43.44** | **70.06** | **19.59** | **43.62** | **69.11** | 15.94 | **41.11** |
| clip_vit | fix | 5.00 | 3.30 | 4.16 | 31.91 | 15.49 | 23.00 | 4.90 | 3.00 | 3.74 | 13.95 | 3.64 | 7.97 | 7.89 | 3.29 | 5.39 |
| | scheduler | **69.80** | **33.90** | **52.79** | **74.83** | **46.64** | **60.99** | **86.70** | **58.60** | **75.01** | **74.35** | **35.67** | **55.54** | **63.17** | **20.87** | **41.05** |
| clip_convnext | fix | 3.10 | 2.50 | 2.82 | 61.76 | 42.13 | 51.54 | 2.80 | 1.60 | 2.23 | 28.89 | 14.33 | 20.92 | 23.90 | 11.33 | 17.14 |
| | scheduler | **81.70** | **50.70** | **67.88** | **81.19** | **52.68** | **67.71** | **90.90** | **74.10** | **84.33** | **79.06** | **42.11** | **61.45** | **70.85** | **25.85** | **48.19** |

Table 1: *Epsilon-Scheduling* **mitigates** *suboptimal transfers* **and consistently improves expected robustness.** Results at moderate perturbation regime ($^4/_{255}$). See Table 2 for $\epsilon_g = {}^8/_{255}$

begins around epoch 25 for $\epsilon_g = {}^4/_{255}$. To the best of our knowledge, the delayed onset of task alignment in robust fine-tuning has not been previously reported.

# 4 Experimental Results

**Pretrained Models and Datasets:** We experiment with six pretrained models—Transformers (*Swin-Base*, *ViT-Base*), Convolutional networks (*ConvNext-Base*, *ResNet50*), and CLIP models (*CLIP-ViT*, *CLIP-ConvNext*)—spanning attention, convolution, supervised, and multi-modal paradigms. Fine-tuning is evaluated on five low-data benchmarks: **CUB-200-2011** (birds), **Stanford Dogs**, **Caltech256**, **Stanford Cars**, and **FGVC-Aircraft**.

*Epsilon-Scheduling* **mitigates** *suboptimal transfer* The results in Table 1 show that while RFT-`fix` often fails to transfer with low clean accuracy, RFT-`scheduler` achieves high clean accuracy for most models. At the same time, it maintains decent adversarial accuracy. For the perturbation target $\epsilon_g = {}^4/_{255}$, while RFT-`fix` sometimes achieves better adversarial accuracy (9 out of 30 configurations), our scheduling strategy always obtains a higher clean and expected accuracy (see also Figure 6 for results aggregated across models). These results show that even at moderate perturbations ($^4/_{255}$), *epsilon-scheduling* prevents the steep degradation incurred by RFT-`fix`, allowing models to retain strong clean performance while achieving improved or similar adversarial accuracy at non-trivial levels. In high perturbation regime ($\epsilon_g = {}^8/_{255}$), transfer fails more often with RFT-`fix` and RFT-`scheduler` becomes the only viable option for robust fine-tuning.

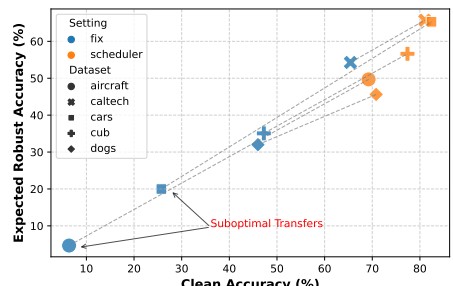

Figure 6: *Epsilon-Scheduling* **mitigates** *suboptimal transfers* **and improves** *expected robustness*. Aggregated results across models from Table 1 ( $\epsilon_g = {}^4/_{255}$).

*Epsilon-Scheduling* **preserves task alignment while improving robustness** Figure 4 shows the evolution of the validation accuracy during training for $\epsilon_g = {}^4/_{255}$. As expected, the standard fine-tuning converges very fast, successfully learning the task with a high clean accuracy. RFT-`fix` negatively affects the clean accuracy and ultimately fails to learn the task effectively. In RFT-`scheduler`, delaying fine-tuning with perturbations helps achieve a high clean accuracy at the level of standard fine-tuning at the early stage. Once RFT starts, around epoch 12, with perturbation strengths above zero, robust accuracy begins to increase. Interestingly, the clean accuracy remains high and relatively stable.

**Limitations & Future Work.** While our study sheds light on the phenomenon of *suboptimal transfer* in RFT and proposes a mitigation via *epsilon-scheduling*, it also opens up several interesting research directions. We leave the study of different schedulers, the mechanistic understanding of *suboptimal transfer*, applications beyond image classification, parameter-efficient fine-tuning, and extensions to other modalities (e.g., language) for future work.

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

## A  Related Work

**Adversarial Robustness in Transfer Learning with Robust-FineTuning**    There are two main ways to achieve adversarial robustness in Transfer Learning: Robust Distillation (Goldblum et al., 2020; Dong et al., 2024) and Robust Fine-Tuning. Previous work on RFT has focused on strategies to preserve the robustness of pretrained models (Liu et al., 2023; Xu et al., 2024; Hua et al., 2024). (Liu et al., 2023) proposed TWINS (TwoWIng NormliSation), a statistics-based fine-tuning framework that employs two neural networks with shared parameters: one maintains the population means and variances of the pretraining data in the batch normalization layers, while the other tracks the statistics of the downstream dataset. AutoLoRA (Xu et al., 2024) shows that there is often a divergence between natural and adversarial gradient directions in RFT and addresses it by disentangling the optimization objectives—using a low-rank LoRA branch for natural objectives and a robust, pretrained feature extractor for adversarial objectives. Hua et al. (2024) showed that linear probing best preserves the robustness of the adversarially pretrained model and proposed RoLi. This strategy initializes the linear classifier head via adversarially trained linear probing before performing RFT. These strategies only consider robust pretrained feature extractors. To the best of our knowledge, this work is the first to propose a method for RFT that directly targets non-robust, pretrained models without assuming robust pretrained features.

**Tuning Perturbation Strength in Adversarial Training**    The idea of tuning or adapting the adversarial perturbation strength $\epsilon$ during training has appeared in various forms across the robustness literature. Early work like Gowal et al. (2018) used a linear ramp-up of $\epsilon$ in the Interval Bound Propagation (IBP) method. Ding et al. (2020) drew a theoretical connection between margin maximization and the loss at the smallest adversarial perturbation, motivating the use of adaptive, sample-specific $\epsilon$ values. Similarly, Balaji et al. (2019) explored instance-wise epsilon selection, though these approaches can be computationally intensive due to per-sample perturbation searches. Ding et al. (2020) additionally introduced PGDLS (PGD with Linear Scaling), which linearly ramps up the perturbation radius during adversarial training and shows little to no improvement at $\epsilon \leq {}^{16}/_{255}$ but only at high $\epsilon = {}^{24}/_{255}$. To better trade off clean and robust accuracy, Chamon & Ribeiro (2020) proposed sampling $\epsilon$ from a Beta distribution. Cai et al. (2018) proposed a curriculum adversarial training scheme that gradually increases the attack steps, which improves performance in combination with batch mixing and quantization. Unlike Pang et al. (2021), which showed that linear $\epsilon$ warmup had a limited effect in ResNets, Debenedetti et al. (2023) showed that it improved both clean and robust accuracy in vision transformers. In contrast to prior works, which have primarily applied perturbation tuning in classical adversarial training from scratch, our study frames *Epsilon-Scheduling* through the lens of transfer learning. In this context, *Epsilon-Scheduling* is not just an optional improvement over standard RFT with a fixed epsilon; rather, it constitutes a dependable alternative when standard RFT fails to transfer, which we show happens when training directly at large $\epsilon$. In addition to previous work, we evaluate performance using a new metric, the *expected robustness*, and show that it is consistently beneficial, regardless of task and architecture, including ResNets.

## B  Additonal Details

**Training Details**    We follow a similar setup described in Hua et al. (2024), using the AdamW optimizer with a cosine learning rate scheduler that includes a warmup period. We select the learning rate and weight decay via hyperparameter optimization (HPO) based on clean accuracy. HPO is performed only for the `fix` setting, and the resulting hyperparameters are reused for the `scheduler` setting to ensure a fair comparison. Adversarial training is performed by minimizing an empirical counterpart of the adversarial risk (Equation 1). More specifically, on a mini-batch $B$ we minimize

$$L_\epsilon(f) = \frac{1}{|B|} \sum_{(x,y) \sim B} \ell_{\text{CE}}(f(\tilde{x}), y)$$

where $\tilde{x}$ is an adversarial example crafted for $x$ using APGD (instead of PGD) with cross-entropy loss as in (Singh et al., 2023; Heuillet et al., 2025), benefiting from APGD's adaptive step size, which removes the need for manual tuning across different perturbation thresholds. The number of APGD steps is 7 for training. As in Heuillet et al. (2025), we train for 50 epochs, and results are reported at the end of training because overfitting of the adversarial accuracy is negligible here (see Figure 4).

| Model | Dataset Metric Setting | Aircraft Clean | Adv. | E. Adv. | Caltech Clean | Adv. | E. Adv. | Cars Clean | Adv. | E. Adv. | Cub Clean | Adv. | E. Adv. | Dogs Clean | Adv. | E. Adv. |
|---|---|---|---|---|---|---|---|---|---|---|---|---|---|---|---|---|
| vit | fix | 3.00 | 2.00 | 2.50 | 44.95 | 19.52 | 31.43 | 3.60 | 2.00 | 2.74 | 17.40 | 2.80 | 8.56 | 8.64 | 2.88 | 5.35 |
|  | scheduler | **57.00** | **6.70** | **27.72** | **72.86** | **26.89** | **49.28** | **68.10** | **9.00** | **35.18** | **64.74** | **9.79** | **33.93** | **56.86** | **5.79** | **25.81** |
| swin | fix | 4.20 | 2.70 | 3.47 | 68.87 | 38.10 | 53.40 | 13.20 | 5.60 | 8.66 | 45.89 | 13.60 | 28.56 | 46.05 | **11.08** | 26.69 |
|  | scheduler | **69.20** | **22.40** | **45.12** | **80.27** | **38.67** | **60.26** | **78.00** | **23.50** | **53.57** | **74.80** | **21.07** | **47.34** | **60.49** | 8.73 | **31.14** |
| convnext | fix | 1.60 | 1.50 | 1.48 | 59.85 | 33.95 | 46.34 | 5.30 | 2.60 | 3.98 | 5.02 | 2.28 | 3.56 | 27.33 | 7.73 | 16.28 |
|  | scheduler | **75.00** | **28.80** | **50.90** | **84.99** | **41.82** | **64.92** | **85.60** | **35.90** | **65.04** | **80.69** | **24.28** | **53.07** | **68.94** | **9.78** | **36.51** |
| r50 | fix | 1.30 | 0.90 | 0.74 | 53.59 | 26.78 | 39.93 | 1.50 | 1.20 | 1.34 | 30.89 | 8.27 | 17.84 | 27.14 | **6.95** | 15.61 |
|  | scheduler | **42.80** | **5.30** | **20.38** | **67.56** | 23.01 | **44.03** | **57.10** | **8.50** | **29.56** | **59.49** | **8.68** | **29.95** | **50.89** | 6.92 | **25.26** |
| clip_vit | fix | 3.60 | 2.20 | 3.05 | 23.02 | 7.29 | 14.52 | 3.00 | 2.50 | 2.73 | 11.11 | 2.30 | 5.73 | 2.20 | 1.38 | 1.77 |
|  | scheduler | **65.80** | **25.40** | **44.84** | **70.68** | **33.70** | **51.67** | **84.70** | **38.60** | **64.47** | **67.64** | **18.05** | **41.79** | **54.28** | **8.94** | **27.78** |
| clip_convnext | fix | 1.80 | 1.30 | 1.62 | 51.94 | 28.37 | 39.44 | 1.30 | 1.10 | 1.25 | 6.37 | 2.30 | 4.05 | 8.36 | 3.97 | 5.98 |
|  | scheduler | **79.20** | **34.50** | **59.09** | **76.53** | **37.20** | **56.83** | **90.00** | **55.20** | **77.14** | **73.58** | **22.75** | **47.77** | **62.67** | **11.36** | **33.85** |

Table 2: *Epsilon-Scheduling* **mitigates *suboptimal transfers* and consistently improves expected robustness in high perturbation regime** ($8/255$). The table shows clean accuracy (**Clean**), adversarial accuracy (**Adv.**), and the *expected adversarial accuracy* (**E. Adv.**). The models are evaluated under a fixed perturbation strength (`fix`) and an *Epsilon-Scheduling* (`scheduler`). See Table 1 for $\epsilon_g = 4/255$

We consider two target evaluation thresholds $\epsilon = 4/255$ (moderate perturbation) and $\epsilon = 8/255$ (high perturbation) as two commonly used evaluation targets on these datasets.

**Evaluation details**    For a given perturbation strength $\epsilon > 0$, the ($L_\infty$-)robust accuracy $\mathrm{Acc}_\epsilon(f)$ of a classifier $f$ is defined as

$$\mathrm{Acc}_\epsilon(f) = \mathbb{E}_{(x,y)\sim D}\mathbf{1}[\forall x'(\|x - x'\|_\infty \leq \epsilon \Rightarrow \arg\max f(x') = y)],$$

where $\mathbf{1}[\phi]$ equals 1 if $\phi$ holds and 0 otherwise. In particular, for $\epsilon = 0$, $\mathrm{Acc}_0(f) = \mathbb{E}_{(x,y)\sim D}\mathbf{1}[\arg\max f(x) = y]$ coincides with the usual clean accuracy of the classifier $f$. This robust accuracy is estimated using the AutoAttack library with the APGD method and 10 steps on a given test dataset. The expected robustness is estimated by using the trapezoidal rule with evaluations made with steps $1/255$, so for example with $\epsilon_g = 4/255$:

$$\mathrm{AUC}_{4/255}(f) = \frac{1}{4}\sum_{i=0}^{3} \frac{\mathrm{Acc}_{\frac{i}{255}}(f) + \mathrm{Acc}_{\frac{i+1}{255}}(f)}{2}.$$

# C    Additional Results

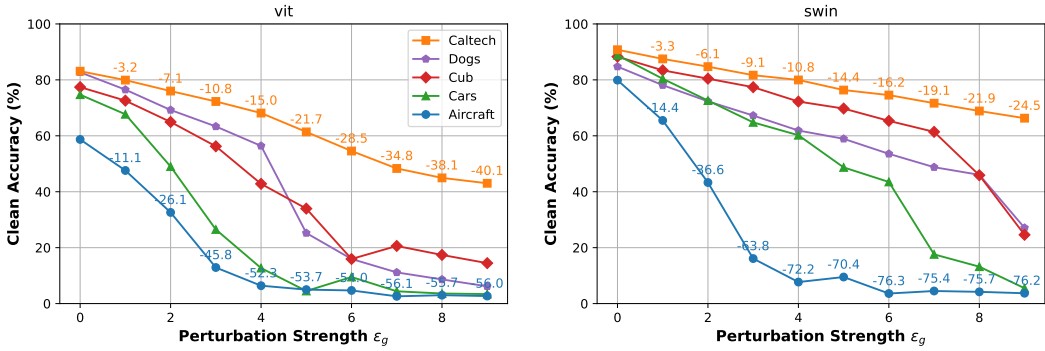

Figure 7: **RFT can lead to *suboptimal transfer* even for small $\epsilon$.** The variation of transfer accuracy with the training perturbation strength $\epsilon_g$ is not always smooth and is highly model- and dataset-dependent.

