# OpenReview forum: "Robust Fine-Tuning from Non-Robust Pretrained Models: Mitigating Suboptimal Transfer With Epsilon-Scheduling"
_NeurIPS.cc/2025/Workshop/Reliable_ML — NeurIPS 2025 - Reliable ML Workshop_

### Official Review · Reviewer_MEHW · 2025-09-15
**Clear phenomenon (suboptimal transfer under RFT‑fix) and a simple, effective scheduling fix with a useful evaluation metric.**

**Rating:** 6
**Confidence:** 4

**Review:**

**Summary**

This paper studies robust fine‑tuning (RFT) from non‑robust pretrained models and finds that training with a fixed adversarial radius $\epsilon_g$ (RFT‑fix) often delays task alignment and yields poor transfer relative to standard fine‑tuning. It proposes Epsilon‑Scheduling as a simple yet effective fix: begin with $\epsilon=0$ for 25% of epochs, increase linearly to $\epsilon_g$ over 50%, then train at $\epsilon_g$ for the final 25%. It further introduces the metric of expected robustness, i.e.  the expectation under uniform distribution U of the accuracy over $[0, \epsilon_g]$. Across six backbone pretrained models (Swin-Base, ViT-Base, ConvNeXt-Base, ResNet50, CLIP‑ViT, CLIP‑ConvNeXt) and five low-data benchmarks (Caltech256, CUB‑200‑2011, Stanford Dogs, Stanford Cars, FGVC-Aircraft), epsilon scheduling mitigates suboptimal transfer and improves expected robustness. Training‑dynamics plots show that epsilon scheduling preserves early clean accuracy while steadily improving robust accuracy.

**Strengths**

**Well‑articulated phenomenon:** The “delayed alignment” under RFT‑fix is clearly documented and is practically important when starting from non‑robust pre-trained models.

**Simple, effective fix:** The 25–50–25 schedule is easy to adopt and consistently improves expected robustness and clean accuracy across architectures/datasets at moderate $\epsilon_g$, as validated by the empirical evidence.

**Metric clarity:** Expected robustness is a helpful summary when clean vs. robust trade‑offs are ambiguous.

**Relevance:** RFT from non‑robust pre-trained models is a common, economical path, and this work surfaces failure modes and a practical remedy aligned with reliable transfer, closely relating to the topic of reliable ML.


**Weaknesses / Limitations**

**Baseline breadth:** The paper cites scheduling/ramp‑up ideas (e.g., PGDLS, IBP) but comparisons are primarily RFT‑fix vs. scheduler. More diverse baselines (e.g., curriculum/adaptive $\epsilon$ strategies) and comparison with existing works would solidify the conclusions.

**Schedule design space:** 25–50–25 is plausible but ad‑hoc. Ablations over warm‑up length, ramp shape, and late‑stage weighting (including cosine/exp schedules or per‑layer ramping) are limited.

**Writing in the experiments part:** Current text analysis on experiments are mostly based on case studies on a small and ad-hoc set of different $\epsilon_g$, though the authors actually perform experiments on a wide range of different $\epsilon_g$. Given the extensive experiments shown in the plotted figures, for more rigorisity, the conclusions can be drawn with some statistical metrics rather than case studies.

**Suggestions for Authors**

**Expand baselines.** Add more existing works and more advanced techniques to the baseline for a comprehensive comparison.

**Schedule ablations.** Vary warm‑up proportion, ramp shapes, per‑batch adaptive $\epsilon$ and show how scheduling interacts with optimizer/lr schedules. An extensive analysis on how to achieve an optimal scheduling pattern will be interesting, adds more interpretability to the current 25-50-25 scheme, and possibly generalizes the current solution.

**Generalization.**  if possible, explore NLP or speech pre-trained models to test cross‑modality claims.

**Enhance rigorisity in experiments.** Substitute case studies with statistical evidence in the explanation of the experiments.

---

### Official Review · Reviewer_dtj1 · 2025-09-19
**Experiments look interesting and the topic is timely. I think I would like to see more novel ideas or additional ablations/insight on RFT.**

**Rating:** 6
**Confidence:** 3

**Review:**

Summary: While many non-robust pre-trained models are available as open-source, it is unclear how to achieve adversarial robustness to adversarial examples while fine-tuning. This work examines standard fine-tuning, existing Robust Fine-Tuning (RFT-fix), and propose a new method which uses the RFT objective with an epsilon schedule which starts at 0 for 25% of epochs, increases linearly from 0 to a hyperparameter $\epsilon_g$, and then remains fixed at $\epsilon_g$. The work shows some initial studies which shed light on a few properties: that in standard RFT-fix, task alignment is delayed until later epochs; perturbation strength in RFT-fix affects transfer accuracy (understandably); and that the RFT-scheduler shows promising results on multiple models and baselines.

Strengths:
- The repeated experiment results across several datasets and models is encouraging and a good sign that RFT-scheduler is more effective than its RFT-fix counterpart for moderate (and high) perturbation levels.
- The experiments and ablations that are currently in the paper are explained and designed well.
- The problem is well-motivated and seems timely and significant.

Weaknesses:
- As stated in the limitations and future work section, the current version of the work is a bit incomplete and would benefit from a more thorough treatment of robust fine-tuning. In its current iteration, the work feels more like an interesting training heuristic. The potential future directions listed are a good place to start. I would also suggest some additional ablation in the low perturbation regime (at which point there may be trade-offs between task accuracy and the robustness).